# Classification Models of Action Research Arm Test Activities in Post-Stroke Patients Based on Human Hand Motion

**DOI:** 10.3390/s22239078

**Published:** 2022-11-23

**Authors:** Jesus Fernando Padilla-Magaña, Esteban Peña-Pitarch

**Affiliations:** Escola Politècnica Superior d’Enginyeria de Manresa (EPSEM), Polytechnic University of Catalonia (UPC), 08242 Manresa, Spain

**Keywords:** machine learning, hand motion, stroke, classification, borderline-SMOTE, finger joints

## Abstract

The Action Research Arm Test (ARAT) presents a ceiling effect that prevents the detection of improvements produced with rehabilitation treatments in stroke patients with mild finger joint impairments. The aim of this study was to develop classification models to predict whether activities with similar ARAT scores were performed by a healthy subject or by a subject post-stroke using the extension and flexion angles of 11 finger joints as features. For this purpose, we used three algorithms: Support Vector Machine (SVM), Random Forest (RF), and K-Nearest Neighbors (KNN). The dataset presented class imbalance, and the classification models presented a low recall, especially in the stroke class. Therefore, we implemented class balance using Borderline-SMOTE. After data balancing the classification models showed significantly higher accuracy, recall, f1-score, and AUC. However, after data balancing, the SVM classifier showed a higher performance with a precision of 98%, a recall of 97.5%, and an AUC of 0.996. The results showed that classification models based on human hand motion features in combination with the oversampling algorithm Borderline-SMOTE achieve higher performance. Furthermore, our study suggests that there are differences in ARAT activities performed between healthy and post-stroke individuals that are not detected by the ARAT scoring process.

## 1. Introduction

Machine learning (ML) is considered a subfield of computer science where knowledge from artificial intelligence and statistics is applied to the generation of computational models. ML algorithms can learn automatically and generate a model from input data without being explicitly programmed to produce a particular output [1]. The learning process is performed through training and the dataset for training is known as training data [2]. The classification algorithms belong to supervised learning. In supervised learning, the algorithm learns through a labeled data set (for example, a set of images labeled as containing a cat or a lion), where each training data sample is presented in the form of an input value with an output label [3]. The algorithm trains a model that, from the input values, can predict the correct response based on the features defined in the process [2]. The classification problems are commonly categorized into binary classification and multi-class classification. The dataset is classified into two classes in binary classification, while in multi-class classification, the given dataset is classified into more than two classes [4]. In recent years classification models have been used in various clinical applications, for example, in medical diagnosis, disease classification, prediction of clinical outcomes, and treatment responses [5,6,7,8]. There are several ML classification algorithms, three of the most widely used in healthcare applications are Support Vector Machine (SVM), Random Forest (RF), and K-Nearest Neighbors (KNN) [9,10,11]. However, in most classification problems in real-life applications, the sample sizes between the different classes are unbalanced [12]. Therefore, the imbalanced data problem is commonly solved with oversampling and undersampling techniques. Synthetic minority oversampling technique (SMOTE), random oversampling (ROS), and random undersampling (RUS) are some of the most commonly used balancing techniques for imbalanced data [13]. The advantage of oversampling over undersampling is that no samples are lost from the original training set since all data from the minority and majority classes are preserved [14]. On the other hand, there is a new oversampling method called Borderline-SMOTE derived from SMOTE that has shown good results in healthcare applications [4,15,16]. SMOTE uses the k-nearest neighbor algorithm to generate new and synthetic data to over sample the minority class [17]. In contrast, Borderline-SMOTE generates the synthetic data around the borderline between the two classes, unlike SMOTE, where synthetic data is created randomly in all the minority samples [18].

Stroke remains one of the leading causes of death and disability in Europe, and projections show that the burden of stroke will not decrease in the next decade or beyond [19]. An important contributing factor to this is that the number of older persons in Europe is rising, with a projected increase of 35% between 2017 and 2050 [20]. After a stroke, one of the main sequelae produced is the loss of mobility in the upper extremities of the human body. Therefore, in the assessment of upper extremity (UE) function it’s important to improve the effectiveness of rehabilitation programs, and the use of standardized outcome measures (OMs) which can lead to more efficient rehabilitation programs for post-stroke patients [21,22]. There are several types of OMs used for evaluating patients with UE disability with good psychometric properties, these are: the Fugl-Meyer assessment (FMA), the Action Research Arm Test (ARAT), the Box and Block test (BBT), the Chedoke Arm and Hand Inventory (CAHAI), the Nine Hole Peg Test (9-HPT), and the Wolf Motor Function Test (WMFT). However, one of the most used OMs by physical therapists and other health care professionals to assess the performance of the UEs in people post-stroke is the ARAT. The ARAT is a measurement tool used to assess UE functional impairments that evaluate 19 movement tasks divided into four subtests (grasp, grip, pinch, and gross arm movement) that assess a patient’s ability to handle objects differing in size, weight, and shape [23]. However, the ARAT, requires a human examiner to transform observations of the patient’s movement into a score [24]. Therefore, the scoring process can be limited to assessing only the quality of performance on each task. In addition, ARAT presents a ceiling effect that prevents detection of improvements produced with rehabilitation treatments in subjects with mild impairments with high ARAT scores [25].

Nevertheless, in recent years several ML models have been developed using hand motion data obtained during the performance of the ARAT. Dutta et al. evaluated grasp abilities and predicted scores in the ARAT test with Support Vector Machine (SVM) algorithms in healthy subjects and post-stroke patients using an instrumented glove composed of six flex sensors, three force sensors, and a motion processing unit [26]. In contrast, Bochniewicz et al. used a Random Forest model to classify UE movement into functional and non-functional, using inertial measurement units (IMU) during the performance of the ARAT [27]. Lum et al. developed several machine learning algorithms K-Nearest Neighbors (KNN), Random Forest, Linear SVM, and SVM Radial Basis Function (RBF) to classify functional and nonfunctional activities using a wrist-worn inertial measurement unit (IMU) during the performance of the ARAT [28]. Moreover, Kanzler et al. predicted outcomes scores with the ARAT, BBT, or NHPT using several machine learning models (decision tree, KNN, linear regression and random forests (RF)) with clinical data, and digital health metrics [29]. However, to the best of our knowledge no machine learning models have been focused on classifying ARAT activities between post-stroke patients with mild impairments and healthy individuals based on hand motion data. Therefore, we present this work, which has two main objectives: (I) Develop classification models to predict whether the ARAT activities were performed by a healthy subject or by a subject post-stroke with good upper extremity functionality, based on the hand motion information obtained with the CyberGlove II^®^ (CyberGlove Systems. LLC, San José, CA, USA). Hence, the high performance of the classification models will demonstrate that there are differences between the activities of healthy and post-stroke subjects that are not detected with the ARAT scoring method. On the other hand, this study has a second objective: (II) Evaluate if data class balancing using the Borderline-SMOTE method allows better performance-classifiers to be obtained. This paper is structured as follows. The data collection and the machine learning architecture development is presented in Section 2. Section 3 presents the results of the performance and comparison of the classification models before and after data class balancing. The discussion is presented in Section 4. Finally, Section 5 concludes this paper.

## 2. Materials and Methods

In this work, we selected three of the most widely used and best-performing classification algorithms in machine learning (ML), which are Support Vector Machine (SVM), Random Forest (RF), and K-Nearest Neighbors (KNN). The development of each of the classification models is presented in this section.

### 2.1. Data Source

The data in this study were obtained from previous studies and comes from the publicly available Finger Joints Angles ARAT database [30]. This section briefly describes the experimental protocol for data acquisition. The extension and flexion angles of eleven finger joints in healthy subjects [31] and post-stroke patients [32] were measured during the performance of sixteen activities of the Action Research Arm Test (ARAT) corresponding to the subtests (grasp, grip, and pinch) using an instrumented glove (Cyberglove Systems LLC; San Jose, CA, USA). The eleven finger joints angles recorded were: thumb carpometacarpal (CMC) joint, thumb, index, middle, ring, and little metacarpophalangeal (MCP) joints, thumb interphalangeal (IP) joint, and index, middle, ring, and little proximal interphalangeal (PIP) joints. For more information about the study protocol, please refer to [31,32]. The information obtained allowed us to construct a dataset composed of the flexion and extension angles of 25 healthy subjects during the performance of 400 ARAT activities and 12 post-stroke patients during the execution of 144 ARAT activities. All the activities were completed with an ARAT score of 2 or 3.

### 2.2. Data Preprocessing

In the dataset each sample was labeled according to the class to which each activity belonged (Control = 0; Stroke = 1). There were 800 cases in class ‘0’ and 288 cases in class ‘1’. Table 1 shows the structure of the dataset used in the model. There were no missing values in the dataset and therefore all 1088 samples were used.

In the dataset we identified the input and output variables, the input was known as feature, and the output was known as response. The dataset has 14 features; the first three correspond to “Activity”, “Subtest”, and “Motion (Extension or Flexion)”, and the remaining eleven correspond to the angles of the finger joints. On the other hand, the response variable is the class to which the subject corresponds {0-Control, 1-Stroke}. Hence, the categorical features in the dataset (x1, x2, x3) were transformed into binary values using one-Hot encoding method. One-Hot encoding transforms a single categorical variable with *p* observations and d distinct values, to d binary variables with *p* observations each. Therefore, each distinct value is converted into a new column and assigned a binary value indicating the true (1) or false (0) value to the column. Importantly, other demographic characteristics were not considered in the ML classifiers because the objective of the study was focused on assessing hand motion information.

### 2.3. Validation Methods

In order to evaluate the overall performance of the classification models, two validation methods were used. The first method used was the hold-out, in which we split our dataset into two parts, a training set and a test set. In each classification model we used 75% of the data as the training set and the remaining 25% as the test set. In addition, the second method used to measure the classification model’s performance was the 5x2cv test. Unlike the common hold-out method, in which we usually split the data set into two parts: a training set and a test set, in the 5x2cv test, we performed five replications of two-fold cross-validation. In each replication, the dataset is divided into two equal sets (50% training and 50% test data).

### 2.4. Tuning Hyperparameters

Hyperparameters are user-adjustable parameters that can vary in quantity from one model to another. There are several computational methods to find the optimal hyperparameters of the model. Two of the most commonly used are GridSearchCV and Randomized Search CV. We decided to use the GridSearchCV; this technique uses all possible permutations of the hyperparameters of a given model. The performance of each model was then evaluated, and the best hyperparameter values were selected. In addition, GridSearchCV has the advantage of performing a K-Fold cross validation, where the number of folds is specified by its cv parameter. If it is not specified, it applied a five-fold cross validation by default. Cross-validation is a technique to identify different problems during model training, such as the occurrence of overfitting. To do this, GridSearchCV will split the training data into training and test partitions to tune the hyperparameters on these data [33].

### 2.5. Classification Metrics

Evaluation metrics for classification models can be applied in two phases. Firstly, in the training phase, to produce a more accurate prediction result in the future evaluation of the classification model. Subsequently, in the testing phase, evaluation metrics are used to measure the efficacy of the classifier when tested on unseen data [34]. Therefore, knowing the different metrics and making the proper selection is crucial to improving the model’s performance. Therefore, to evaluate the performance of the classification models, we used the following metrics in this work: confusion matrix (not a metric but fundamental to the others), precision, accuracy, recall, F1-score, area under the receiver operating characteristics (AU-ROC), and the classification report. The formulas corresponding to the evaluation metrics are shown in Table 2.

#### Confusion Matrix

A confusion matrix is a useful tool for analyzing the performance of classification models when tested on unseen data. A confusion matrix is a cross table of true labels versus model predictions. Each row of the confusion matrix represents instances of an actual class, and each column represents instances of a predicted class [34]. Typically, it is used for binary classification problems but can also be applied to multi-class classification problems. In Figure 1 a binary confusion matrix of 2 × 2 is shown.

### 2.6. Over-Sampling Data

In this work, the classes in the data set are slightly unbalanced. Therefore, the imbalanced data problem is solved with oversampling or undersampling techniques. The advantage of oversampling over undersampling is that no samples are lost from the original training set, since all data from the minority and majority classes are preserved [14]. However, in a large dataset, the time and memory consumption could be very large and costly in oversampling. Since the dataset in our study is not huge and the imbalance is mild, we do not face this problem. Therefore, we selected an oversampling technique. We decided to use the Borderline-SMOTE algorithm motivated by the results in studies of arrhythmia detection [15], estimation of brain metastasis [16] and emotion recognition [4].

Borderline-SMOTE is an algorithm derived from SMOTE (Synthetic Minority Over-sampling Technique). Borderline-SMOTE generates the synthetic data around the borderline between the two classes [18]. The procedure is as follows: First, we calculate the nearest neighbors in the minority class N in all the training set samples. Next, we identified the nearest neighbors; if the majority correspond to the majority class, the samples are put in a set called Danger. The samples in Danger correspond to the borderline data of the minority class. Then, we selected a random N nearest neighbors for each sample in Danger to create the synthetic data. Therefore, we calculate the distance between the sample and its N nearest neighbors and multiply by a random number between 0 and 1. Finally, the synthetic samples of the minority class are generated:$$Synthetic=p_{j}+r_{j}\times dif_{j},\;j=1,2,\ldots ,s\;$$
where *p_j_* represents the samples in Danger, *r_j_* represents a random number between [0, 1], and *dif_j_* represents the distance between the samples and the N nearest neighbor.

### 2.7. Statistical Analysis

The three classification models (RF, SVM, KNN) were compared to determine which has the best performance. First, we presented the results obtained with the GridSearchcv technique on the three classification models in different evaluation metrics such as Accuracy, Precision, Recall, F1-score, and AUC-ROC using the hold-out method. Next, the overall performance of the models were obtained and compared with the 5x2cv combined F test [35] using the MLxtend library by Sebastian Raschka [36], which provided the f-statistic and *p*-value. Subsequently, we evaluated the performance of the three classification models using the hold-out method with several evaluation metrics after balancing the data classes with the technique Borderline-SMOTE. Then, the evaluation metrics of the three models after data balancing were obtained and compared using the 5x2cv combined F test. Finally, each classification model was compared before balancing and after balancing using the 5x2cv t paired test. The statistical analysis was conducted using the software Anaconda (Anaconda Inc, Austin, TX, USA) with Python 3.9. A *p*-value of less than 0.05 was considered statistically significant for all the statistical analyses.

## 3. Results

In this section, the results of the performance and the comparison of the classification models Random Forest (RF), K-nearest Neighbor (KNN), and Support Vector Machine (SVM) are presented.

### 3.1. Hyperparameters Selection

The GridSearchCV technique was applied to each of the classification algorithms (RF, KNN, SVM) developed in this work using a cross validation fold value of five. The best performing hyperparameter values obtained in each of the classification models were as follows.

SVM: [‘C’: 10, ‘gamma’: 0.1, ‘kernel’: ‘rbf’]KNN: [‘leaf_size’: 20, ‘metric’: ‘minkowski’, ‘n_neighbors’: 10, ‘p’: 3, ‘weights’: ‘distance’]RF: [‘max_depth’: 50, ‘min_samples_leaf’: 1, ‘min_samples_split’: 3, ‘n_estimators’: 500]

### 3.2. Classification Models with GridSearchCV

The hyperparameter values obtained were used to evaluate each classifier in the prediction of results in the test set and the results were as follows.

#### 3.2.1. Random Forest

The RF classifier showed an accuracy of 93% and a high precision of 96.5%. In contrast, the recall of 76.4% and the f1-score of 85.3% were low. On the other hand, the classification report presented in Table 3 shows that the recall and the f1-score values were higher in the control class but were lower in the stroke class. In contrast, the precision was higher in the stroke class, as is shown in the confusion matrix in Figure 2.

#### 3.2.2. K-nearest Neighbor

The KNN classifier presented a high precision of 95.3% and an accuracy of 87.9%. In contrast, the recall of 56.9 % and the f1-score of 71% were low. Table 4 shows that the recall and f1-score were higher in the control class. While the precision was low in the control class as is shown in Figure 3.

#### 3.2.3. Support Vector Machine

The SVM classifier showed a high precision of 98.3% and a high accuracy of 94.5%. In contrast, the SVM classifier showed a recall of 80.5% and an f1-score of 88.5%. However, the classification report in Table 5 showed high values in precision, recall, and f1-score in the control class and in the precision of the stroke class, as is shown in the confusion matrix in Figure 4.

### 3.3. Performance Comparison of Classification Models

Table 6 shows the mean values and standard deviation of several evaluation metrics for the RF, SVM, and KNN classifiers. The results showed that the three models have similar precision, and no significant differences were found among the three classifiers (*p* > 0.05). The SVM classifier showed significantly higher accuracy and f1-score than the KNN classifier. In contrast, no significant differences (*p* > 0.05) were found in accuracy and f1-score between the SVM and RF classifiers. In addition, the SVM classifier showed a significantly higher recall and AUC than the RF and the KNN classifiers.

### 3.4. Borderline-SMOTE Data Balancing

The dataset in this work presented a mild case of imbalanced data between the two classes (control and stroke) as is shown in Figure 5. The results presented earlier in this work showed a high accuracy in the three classifiers. However, the accuracy metric is not a good indicator when there are imbalanced classes, as in this case. In contrast, the classifiers KNN and RF showed a lower recall especially in the classification of subjects of the control class. Therefore, to optimize the performance of the classifiers, we decided to use the Borderline-SMOTE algorithm for data oversampling. In contrast, Figure 6 shows the classes after the data balancing with Borderline-SMOTE.

### 3.5. Classification Models with Borderline-SMOTE

The three classification models showed an improvement in the classification of both classes (control and stroke), as is shown in Table 7 after data balancing with Borderline-SMOTE. Previously, the classification models showed difficulty in classifying subjects with stroke due to the class imbalance. The improvement was remarkable, particularly in the metrics of recall and f1-score. The overall performance of the classification models in the test set was evaluated with the following evaluation metrics: accuracy, precision, recall, F1-score, and AUC-ROC. The results were as follows: The SVM classifier showed a precision of 98%, while the RF classifier showed a precision of 96.4%, and the KNN classifier showed a precision of 86.3%. On the other hand, the KNN classifier presented a recall value of 98%, but analyzing the classification report in Table 7 we observed a low recall value of 84% in the control class. In contrast, the SVM classifier presented a recall value of 97.5% and the RF presented a recall of 94% and both classifiers had a uniform recall value in the two classes. In addition, the SVM classifier showed the highest f1-score of 97.7 %, while the RF classifier showed a f1-score of 95.2% and the KNN classifier showed a f1-score of 91.8%. Finally, the AUC-ROC of the three classifiers is shown in Figure 7.

### 3.6. Performance Comparison between Classifiers after Borderline-SMOTE

Table 8 shows the mean values and standard deviation of several evaluation metrics for the RF, SVM, and KNN classifiers using Borderline-SMOTE. As can be seen, the three classification models have a similar and consistent performance. The results showed no significant differences (*p* > 0.05) in accuracy, recall, and f1-score among the three classifiers. However, the RF classifier showed significantly higher precision than the KNN classifier. In contrast, the RF and the SVM classifiers showed a significantly higher AUC than the KNN classifier.

### 3.7. Performance Comparison of the Classifiers before and after Data Balancing

In general, the three classification models RF, KNN, SVM showed an improvement after the data balancing process using the Borderline-SMOTE technique in several metrics. In the statistical comparison of the classifiers with unbalanced data and with data balancing, the following results were obtained. In precision metrics, no significant differences were found in the RF and SVM classifiers before data balancing and after data balancing. In contrast, the KNN classifier after data balancing showed significantly lower precision than before balancing, as is shown in Figure 8. On the other hand, the RF, SVM, and KNN classifiers showed significantly higher accuracy, recall, f1-score, and AUC after data balancing, as is shown in Figure 8, Figure 9, Figure 10, Figure 11 and Figure 12.

## 4. Discussion

In this work, three binary classification models were developed using the following algorithms: Support Vector Machine (SVM), Random Forest (RF), and K-nearest neighbor (KNN). We used as features, the angles of flexion and extension of 11 finger joints during the performance of the activities of the Action Research Arm Test (ARAT) to classify activities between two classes: healthy subject (0 = Control) and post-stroke patients (1 = Stroke). Importantly, the activities included in the dataset of the stroke group obtained a score of 2–3 on the ARAT. While in all the activities included in the dataset of the control group, a score of 3 was obtained. Therefore, based on the ARAT score all tasks were completed and there is not much difference between the tasks in one group and the other. In this way, using completed tasks based on the ARAT score, the performance of the classifier is evaluated.

The result showed that the SVM classifier had the best performance in the test set before data balancing with a precision of 98.3%, an accuracy of 94.5 %, a recall of 80.5%, f1-score of 88.5% and an AUC of 0.989. In addition, the recall and AUC were significantly higher than the RF and KNN classifiers. However, the recall values in the three classifiers were low especially in the stroke group. The lower recall showed that the model was classifying stroke patients as healthy subjects as is shown in the confusion matrices in Figure 2, Figure 3 and Figure 4. Besides, we detected a mild imbalance between classes, so we decided to use the oversampling technique Borderline-SMOTE. The results showed that after Borderline-SMOTE the three classifiers showed significantly higher accuracy, recall, f1-score, and AUC. In the recall, there was an increase in the RF (17.6%), SVM (17%), and KNN (41.1%). However, the KNN classifier showed a recall of 84% in the control class while the RF and SVM classifiers had a balanced recall in both classes. In addition, the precision of the KNN was significantly lower after data balancing. Finally, all the classification models showed an AUC > 0.95. In fact, the RF and SVM showed the best performance after data balancing and no significant differences were found in any metric between the two classifiers. However, the SVM showed a high accuracy, recall and f1-score and therefore a more balanced performance. Hence, The SVM was the model with the best performance after data balancing with Borderline-SMOTE and before data balancing.

The results in our study showed that after data balancing using Borderline-SMOTE the classification models showed an improved performance as in other research. Reddy et al. applied the Borderline-SMOTE algorithm on convolutional neural networks (CNN) to detect arrhythmias using electrocardiogram signals. Their results showed a significantly higher f1-score and accuracy after the use of Borderline-SMOTE [15]. The results of our work were similar, where the three classification models obtained significantly higher accuracy, recall and f1-score after using the Borderline-SMOTE technique. On the other hand, Chang et al. presented a study for emotion recognition with electroencephalogram (EEG) signals using data augmentation with the Borderline-SMOTE method [4]. They compared traditional machine learning methods, and their results showed that SVM and XGBoost had better performance in average accuracy and average macro f1-score than decision tree and KNN models. Our work obtained similar results where the SVM algorithm showed higher accuracy and f1-score than the KNN and RF models. However, our study used a binary classification analysis, and in the study of Chang et al., a multiclass classification method was used.

In addition, there are few studies that have developed machine learning models with data obtained from the ARAT test. Dutta et al. developed an SVM classifier with an accuracy of 92% to predict ARAT scores in patients with different degrees of disability. They used a glove with six flex sensor and three force sensors and a motion processing unit [26]. In contrast, Dutta et al. predicted ARAT scores (0, 1, 2, 3) using a multiclass classification model and presented problems in classifying classes 0 and 2. On the other hand, Rheme et al. used an SVM model to predict good and poor motor outcomes of stroke patients based on the ARAT score, grip force index, and magnetic resonance imaging (fMRI) using a SVM classifier with an accuracy of 76% [37]. However, the outcome of patients with initially moderate impairment could not be predicted with the information of the ARAT score and the grip force test. The results of the above studies demonstrate that it is difficult to predict in post-stroke subjects with moderate impairments based on the ARAT scoring process which confirms what was established in our study. Therefore, the difference with our work is that our study was not based on the ARAT score, but on the information of eleven finger joints angles obtained with the human hand motion system composed by a data glove with 18 flexion sensors during the performance of the ARAT. In addition, in our study we obtained an accuracy of 97.8% with the SVM model, 97.1% with the RF model, and 94.8% with the KNN model classifying ARAT activities. Therefore, our classification models presented better results than those presented in the studies by Dutta et al. and Rheme et al. Furthermore, to our knowledge, there are no previous studies that have developed machine learning models to classify ARAT activities using the range of motion of the finger joints as features and, therefore, a direct comparison with other studies will be biased as most of these use demographic characteristics and scores on several outcome measures as features.

For these reasons, our results demonstrate that using human hand motion information allows the development of high-performance Machine Learning models (SVM, KNN, RF) for classifying ARAT activities. However, to achieve these results it was necessary to balance the classes using the Borderline-SMOTE algorithm. Importantly, we demonstrate that there are differences not detected by the ARAT scoring process that are limited only to evaluate the quality performance. Therefore, these results are of clinical relevance for physiotherapists and other health care professionals who can use a classification model for the detection of finger joint impairments not only in people post-stroke but also after surgical procedures, hand injuries and other hand disorders.

Nevertheless, the present study had some limitations. We were limited to the use of traditional machine learning classifiers, but the use of Ensemble Machine Learning methods has shown very good results in clinical studies [38,39], so that in future work we could implement an ensemble classifier and compare the results. In addition, deep learning has also been used for classification problems in the area of healthcare, especially Convolutional neural networks (CNNs), showing good results in combination with Borderline-SMOTE [4,15]. Moreover, Fraiwan et al. used deep transfer learning to detect scoliosis and spondylolisthesis [40]. Therefore, the use of deep learning should also be considered for future studies. On the other hand, other research to improve the performance of disease predictions uses the development of a classification algorithm based on a multi-level iterative influence measure that would be interesting to use in future research [41]. Another limitation we had is that it was not possible to access patient demographic data. This would have allowed us to evaluate the impact of these features on the classification algorithm. Finally, we limited ourselves to evaluating patients with good upper extremity function according to the ARAT score. Therefore, it would be interesting to evaluate subjects with different degrees of upper extremity impairment to perform a classification model to predict ARAT scores based on information from the eleven finger joints. On the other hand, it is essential to consider the recall results since the activities of post-stroke patients identified as the control group could have been performed in the same way as a healthy person. Therefore, the bias would not be in the classifier model, but in the fact that there were no significant differences in certain activities between post-stroke patients and healthy subjects.

## 5. Conclusions

In this study, we present the novel development of classification models based on human hand motion features in combination with the oversampling algorithm Borderline-SMOTE. The classifiers with unbalanced data showed a low recall and f1-score especially in the stroke class, after the implementation of Borderline-SMOTE the three classifiers showed a significantly higher accuracy, recall, f1-score, and AUC. However, the SVM classifier showed the higher performance with a precision of 98%, a recall of 97.5% and an AUC of 0.996 after data balancing. Therefore, the results showed that the classification models using Borderline-SMOTE achieve a higher performance. In addition, the high performance of the classifiers showed that there are differences between the activities performed between healthy and post-stroke individuals that are not detected by the ARAT scoring process. Regardless, the recall results can show activities in which people from both classes performed equally well. Furthermore, the classification model based on hand motion information can be used in future work for the detection of finger joint impairments not only in people post-stroke but also after surgical procedures, hand injuries, and other hand disorders.

## Figures and Tables

**Figure 1 sensors-22-09078-f001:**
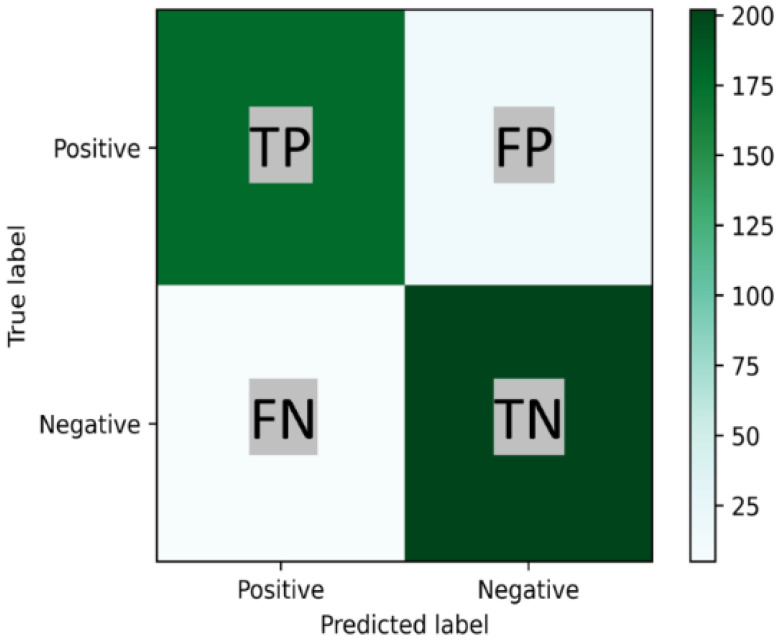
Confusion Matrix Binary Classification; TP represents the number of positive and TN the negative instances that are correctly classified. For its part, FP and FN represent the number of misclassified negative and positive instances, respectively.

**Figure 2 sensors-22-09078-f002:**
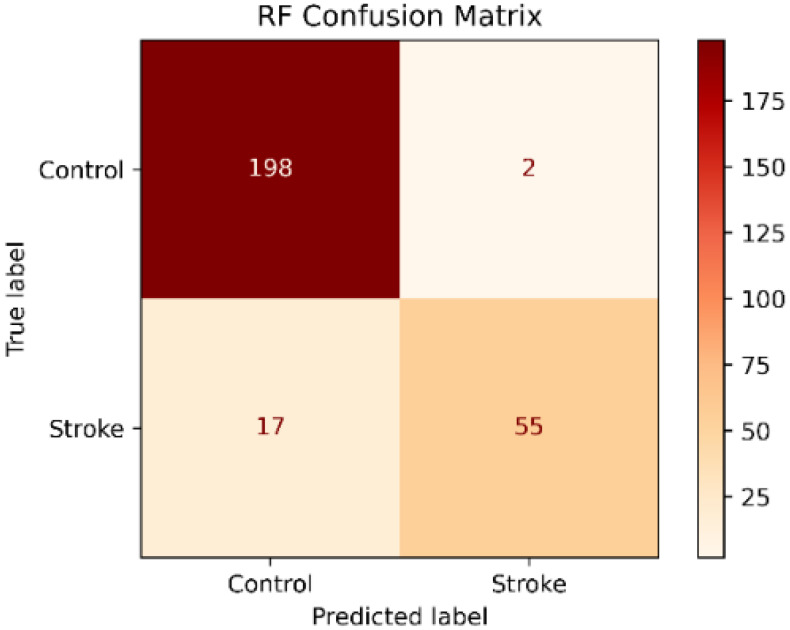
Random Forest Confusion Matrix.

**Figure 3 sensors-22-09078-f003:**
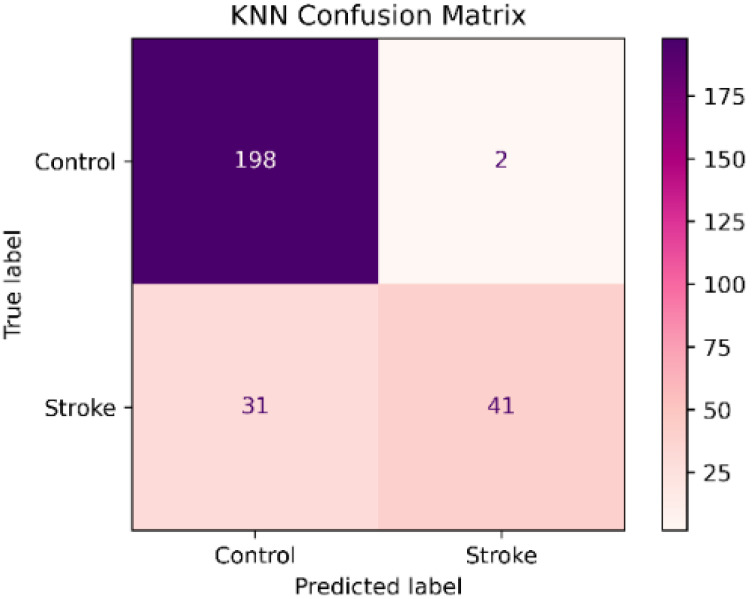
K-nearest Neighbor Confusion Matrix.

**Figure 4 sensors-22-09078-f004:**
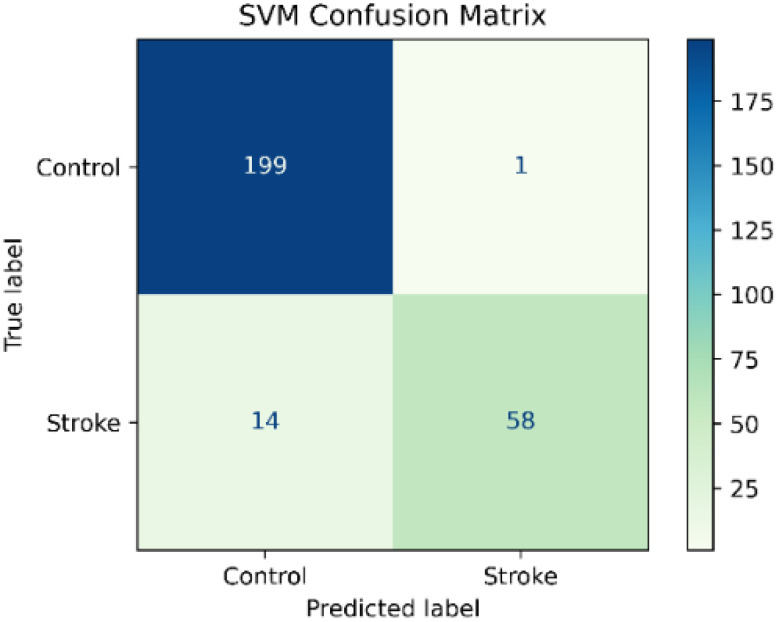
Support Vector Machine Confusion Matrix.

**Figure 5 sensors-22-09078-f005:**
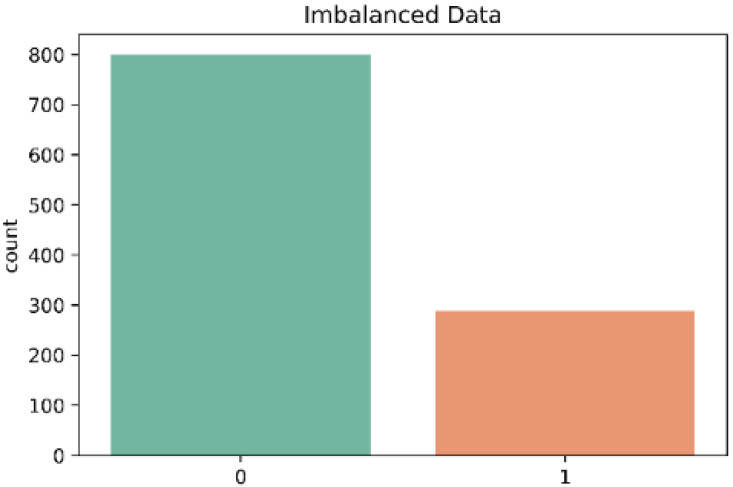
Distribution of each class in the dataset; 0 = Control and 1 = Stroke.

**Figure 6 sensors-22-09078-f006:**
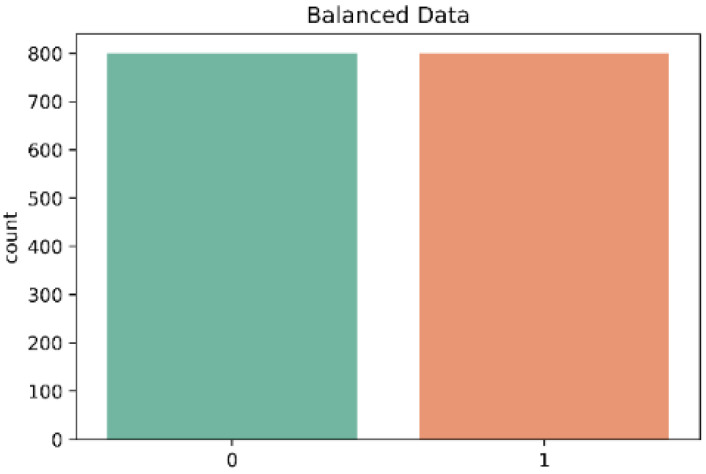
Distribution of each class in the dataset after Borderline-SMOTE; 0 = Control and 1 = Stroke.

**Figure 7 sensors-22-09078-f007:**
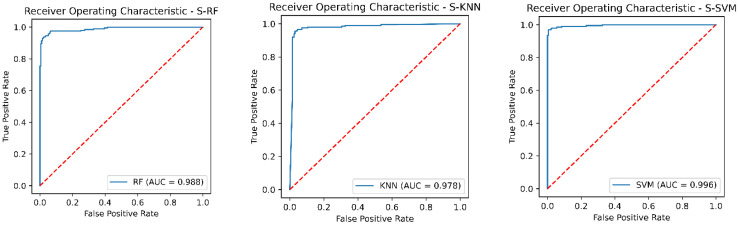
The receiver operating characteristic curves (ROC) of the three classification models with Borderline-SMOTE; Subfigure (**left**) shows the ROC of the classifier Random Forest; Subfigure (**center**) shows the ROC of the classifier Support Vector Machine; Subfigure (**right**) shows the ROC of the classifier K-nearest neighbor.

**Figure 8 sensors-22-09078-f008:**
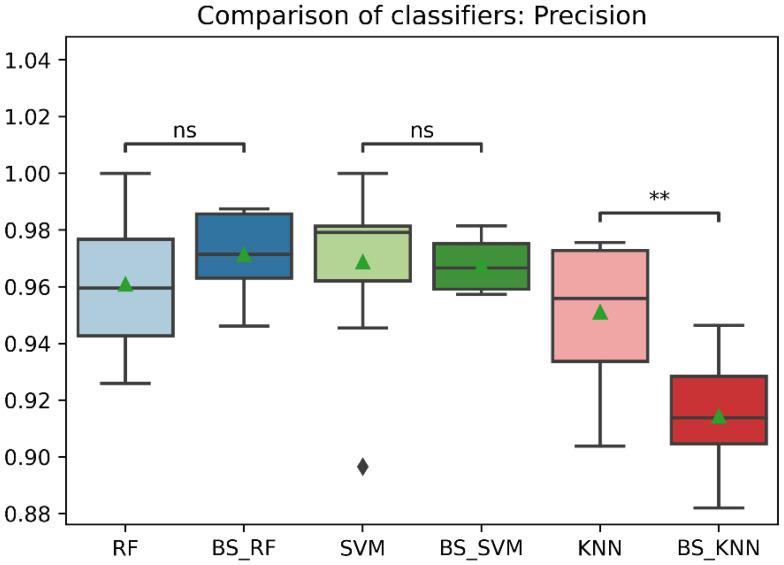
Comparison of precision between classification models with imbalanced and oversampled data. RF = Random Forest; BS_RF = Borderline-SMOTE in Random Forest; SVM = Support Vector Machine; BS_SVM = Borderline-SMOTE in Support Vector Machine; KNN = K-nearest Neighbors; BS_ KNN = Borderline-SMOTE in K-nearest Neighbors; ns: *p* > 0.05; **: *p* ≤ 0.01.

**Figure 9 sensors-22-09078-f009:**
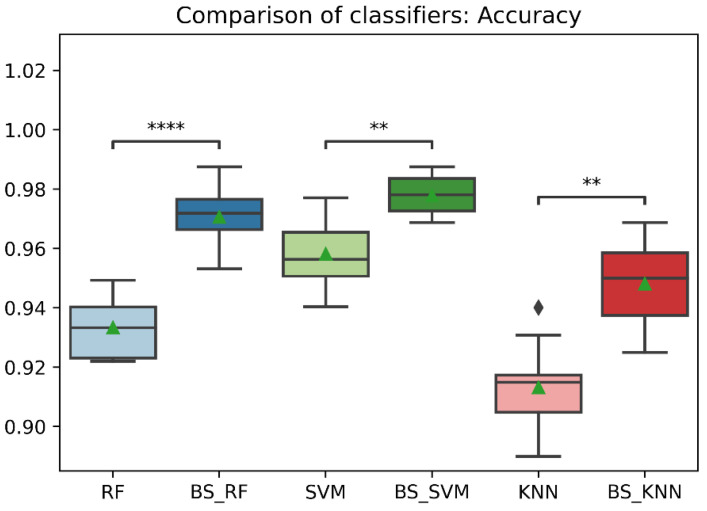
Comparison of accuracy between classification models with imbalanced and oversampled data. RF = Random Forest; BS_RF = Borderline-SMOTE Random Forest; SVM = Support Vector Machine; BS_SVM = Borderline-SMOTE Support Vector Machine; KNN = K-nearest Neighbors; BS_ KNN = Borderline-SMOTE K-nearest Neighbors; **: *p* ≤ 0.01; ****: *p* ≤ 0.0001.

**Figure 10 sensors-22-09078-f010:**
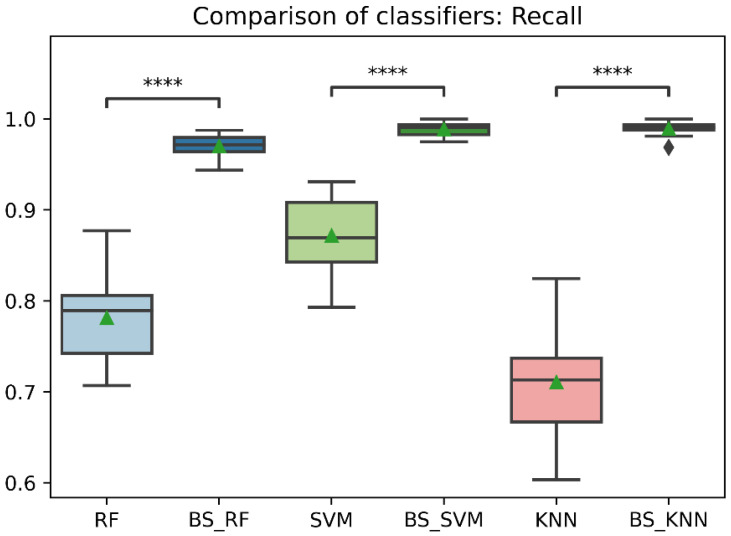
Comparison of recall between classification models with imbalanced and oversampled data. RF = Random Forest; BS_RF = Borderline-SMOTE Random Forest; SVM = Support Vector Machine; BS_SVM = Borderline-SMOTE Support Vector Machine; KNN = K-nearest Neighbors; BS_ KNN = Borderline-SMOTE K-nearest Neighbors; ****: *p* ≤ 0.0001.

**Figure 11 sensors-22-09078-f011:**
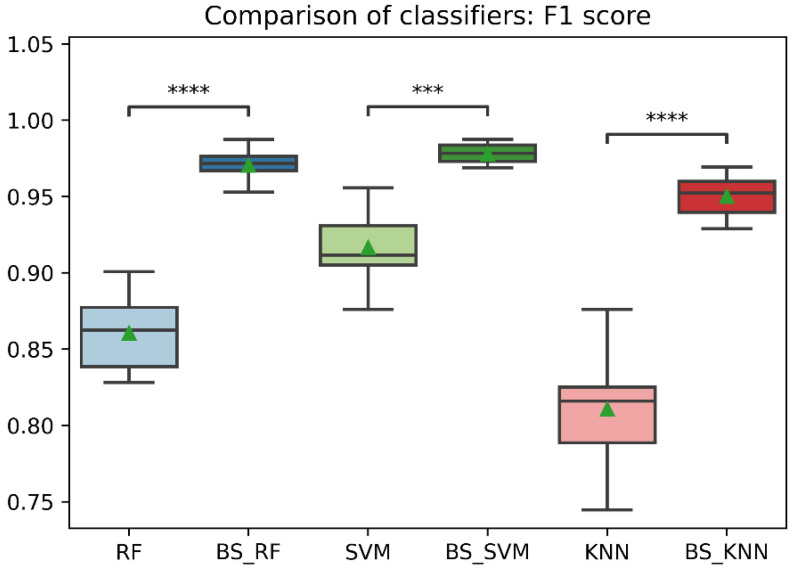
Comparison of F1-score between classification models with imbalanced and oversampled data. RF = Random Forest; BS_RF = Borderline-SMOTE Random Forest; SVM = Support Vector Machine; BS_SVM = Borderline-SMOTE Support Vector Machine; KNN = K-nearest Neighbors; BS_ KNN = Borderline-SMOTE K-nearest Neighbors; ***: *p* ≤ 0.001; ****: *p* ≤ 0.0001.

**Figure 12 sensors-22-09078-f012:**
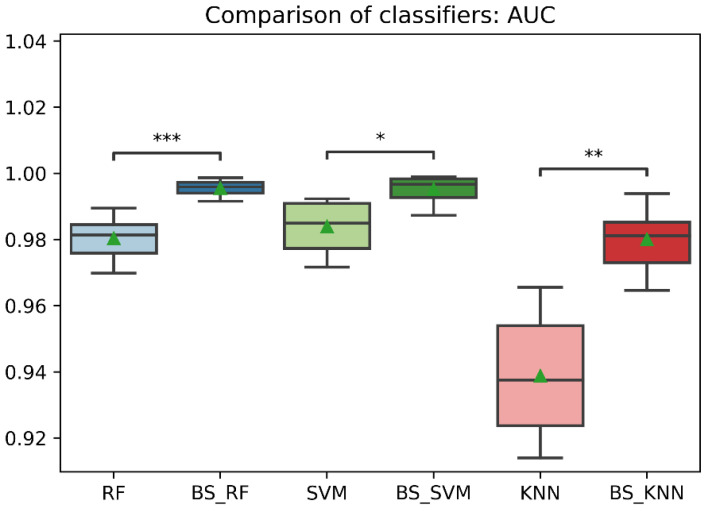
Comparison of AUC between classification models with imbalanced and oversampled data. RF = Random Forest; BS_RF = Borderline-SMOTE Random Forest; SVM = Support Vector Machine; BS_SVM = Borderline-SMOTE Support Vector Machine; KNN = K-nearest Neighbors; BS_ KNN = Borderline-SMOTE K-nearest Neighbors; *: *p* ≤ 0.05; **: *p* ≤ 0.01; ***: *p* ≤ 0.001.

**Table 1 sensors-22-09078-t001:** Dataset variables.

y1	x1	x2	x3	x4 (°)	x5 (°)	x6 (°)	x7 (°)	x8 (°)	x9 (°)	x10 (°)	x11 (°)	x12 (°)	x13 (°)	x14 (°)
0	Test 01	Grasp	Extension	4.47	13.51	1.73	−16.75	7.72	8.02	3.93	−17.50	3.16	8.64	2.13
1	Test 12	Grip	Flexion	18.42	17.87	23.39	28.44	9.41	27.20	10.76	11.30	13.52	21.26	4.42
0	Test 13	Grasp	Extension	6.70	13.78	0.65	20.91	13.05	22.22	11.41	13.85	12.03	20.11	3.68
1	Test 04	Pinch	Flexion	13.85	15.30	7.08	17.12	10.53	19.67	8.00	13.23	10.31	18.02	4.89

y1 = response of the dataset; x1–x14 = features of the dataset; x1 = activity, x2 = subtest; x3 = type of motion; x4 = thumb CMC, x5 = thumb MCP, x6 = thumb IP, x7 = index MCP, x8 = index PIP, x9 = middle MCP, x10 = middle PIP, x11 = ring MCP, x12 = ring PIP, x13 = little MCP, x14 = little PIP; ° = degree.

**Table 2 sensors-22-09078-t002:** Formulas of evaluation metrics.

Metric	Formula	Description
Accuracy	$\frac{TP+TN}{TP+FP+TN+FN}$	Accuracy measures the ratio of the number of correct predictions over the total number of predictions. Therefore, accuracy measures how often the classifier correctly predicts.
Precision	$\frac{TP}{TP+FP}$	Precision measures the positive instances that are correctly predicted from the total predicted instances in a positive class.
Recall	$\frac{TP}{TP+FN}$	Precision measures the positive instances that are correctly predicted from the total predicted instances in a positive class.
F1-score	$2\times \frac{Recall\times Precision}{Recall+Precision}$	F-score or F1-score evaluates the recall and precision at the same time. Therefore, F-score is maximum if the recall is equal to the precision.

**Table 3 sensors-22-09078-t003:** Random Forest model classification report.

Classes	Precision	Recall	F1-Score	Support
Control	0.92	0.99	0.95	200
Stroke	0.96	0.76	0.85	72

**Table 4 sensors-22-09078-t004:** K-nearest neighbor model classification report.

Classes	Precision	Recall	F1-Score	Support
Control	0.86	0.99	0.92	200
Stroke	0.95	0.57	0.71	72

**Table 5 sensors-22-09078-t005:** Support Vector Machine model classification report.

Classes	Precision	Recall	F1-Score	Support
Control	0.93	0.99	0.96	200
Stroke	0.98	0.81	0.89	72

**Table 6 sensors-22-09078-t006:** Comparison of classification models in different evaluation metrics.

Evaluation Metric	RF	SVM	KNN	RF-VM		RF-KNN		SVM-KNN	
	Mean ± SD	Mean ± SD	Mean ± SD	*f*	*p*	*f*	*p*	*f*	*p*
Precision	0.961 ± 0.02	0.969 ± 0.03	0.951 ± 0.02	1.459	0.355	1.788	0.271	1.421	0.366
Accuracy	0.933 ± 0.01	0.958 ± 0.01	0.913 ± 0.01	2.917	0.124	2.2	0.199	26.388	0.001 ***
Recall	0.781 ± 0.04	0.872 ± 0.04	0.710 ± 0.07	8.959	0.013 *	1.089	0.513	45.417	0.00 ***
F1-score	0.861 ± 0.02	0.917 ± 0.02	0.811 ± 0.04	3.789	0.077	1.7	0.29	36.179	0.00 ***
AUC	0.980 ± 0.01	0.984 ± 0.03	0.939 ± 0.02	5.445	0.038 *	8.15	0.016 **	11.754	0.007 **

RF = Random Forest; SVM = Support Vector Machine; KNN = K-nearest Neighbors; SD = standard deviation; *f = f* statistic; *: *p* ≤ 0.05; **: *p* ≤= 0.01; ***: *p* ≤= 0.001.

**Table 7 sensors-22-09078-t007:** Classification Report of the three classification models after data balancing (Borderline-SMOTE).

Evaluation Metric	RF		SVM		KNN	
	Control	Stroke	Control	Stroke	Control	Stroke
Precision	0.94	0.96	0.98	0.86	0.98	0.98
Recall	0.96	0.94	0.84	0.98	0.98	0.97
F1-score	0.95	0.95	0.91	0.92	0.98	0.98
Support	200	200	200	200	200	200
Accuracy	0.95		0.98		0.91	

RF = Random Forest; SVM = Support Vector Machine; KNN = K-nearest Neighbors.

**Table 8 sensors-22-09078-t008:** Comparison of classification models after Borderline-SMOTE in different evaluation metrics.

Evaluation Metric	BS_RF	BS_SVM	BS_KNN	BS_RF-BS_SVM		BS_RF-KNN		BS_SVM-KNN	
	Mean ± SD	Mean ± SD	Mean ± SD	*f*	*p*	*f*	*p*	*f*	*p*
Precision	0.971 ± 0.014	0.968 ± 0.008	0.914 ± 0.021	0.551	0.803	6.083	0.03 *	4.049	0.068
Accuracy	0.971 ± 0.010	0.978 ± 0.006	0.948 ± 0.014	1.691	0.293	2.81	0.133	3.88	0.074
Recall	0.970 ± 0.014	0.989 ± 0.007	0.989 ± 0.009	3.403	0.094	3.8	0.077	2.765	0.137
F1-score	0.971 ± 0.010	0.978 ± 0.006	0.950 ± 0.013	1.833	0.261	2.525	0.159	3.668	0.082
AUC	0.996 ± 0.002	0.995 ± 0.004	0.980 ± 0.009	4.074	0.067	7.457	0.019 *	7.098	0.022 *

BS_RF = Borderline-SMOTE in Random Forest; BS_SVM = Borderline-SMOTE in Support Vector Machine; BS_ KNN = Borderline-SMOTE in K-nearest Neighbors; SD = standard deviation; *f* = *f* statistic; *: *p* ≤ 0.05.

## Data Availability

The datasets generated and analyzed during the current study are available at this repository Padilla, Jesus (2022): Finger Joints Angles ARAT. Figshare. Dataset. https://doi.org/10.6084/m9.figshare.19467269.v1 (accessed on 4 March 2022).

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
