# Peer review of "Classification Models of Action Research Arm Test Activities in Post-Stroke Patients Based on Human Hand Motion"

_sensors, 2022, doi:10.3390/s22239078_

Round 1
Reviewer 1 Report
- Tuning the hyper paramteres is different from cross-validations. Cross validations is necessary for the reporting of the results. Moreover it is very important to ensure the transparency and reproducibility of the research via detailed description of the used methods. The evaluation of the computational pipeline should be based on an independent data set. In case of physiological time series one of the accepted options is subject-wise leave-one out validation, as it is resembling the real world setup.
- The novelty and contribution of the paper need to be firmly estalbished and compared to exisitng works.
- The table of abbreviations as required by the journal template is missing.
- receiver operating characteristic curves (ROC) and other abbreviations are defined many times. Only define at first reference.
Author Response
"Please see the attachment."

Reviewer 2 Report
In this article, the authors proposed Classification Models of Action Research Arm Test activities in 2 Post-stroke patients based on Human Hand Motion. The novelty in the paper is good and results are satisfactory. There are some suggestions which authors need to include in their manuscript.
1. Compare the results with existing state-of-art techniques on the same dataset.
2. Discuss how the categorical features(x1,x2,x3) are handled in the simulations.
3. Some of the abbreviations are repeated. Correct them.
Author Response
"Please see the attachment."

Reviewer 3 Report
The authors interest in their work aim to develop classification models to predict whether activities with similar ARAT scores were performed by a healthy subject or by a subject post-stroke using the extension and flexion angles of 11 finger joints as features. To achieve this classification the authors used three algorithms to compare the results : Support Vector Machine (SVM), Random Forest (RF), and K-Nearest Neighbors (KNN).
The paper well-constructed.
The authors presented a good overview for the state of the art.
The contributions of the research are presented.
The authors presented a good quality of analysis.
The paper in current state considered for publication, yet there are some changes need to be carried out for final version.
· I recommend the authors to add another keyword, generally in research paper at least we use six keywords.
· At the end of introduction section, it should contain an outlines section.
· I recommend the authors to revise the citation styles in this journal: [5]-[8].
· It is better to change name of section 2 to Materials and methods.
· I recommend the authors to avoid make two consecutive titles, it is better to add small introduction.
· The authors should some examples how to use one-Hot encoding method for the used dataset, and they should present some information about this method.
Author Response
"Please see the attachment."

Round 2
Reviewer 1 Report
The authors answered my comments. It would be great if the authors included some relevant classification reference relating to recent trends (e.g., deep learning), see
Using deep transfer learning to detect scoliosis and spondylolisthesis from x-ray images. PLOS ONE 17(5): e0267851. https://doi.org/10.1371/journal.pone.0267851
and
Ananthajothi, K., Subramaniam, M. Multi level incremental influence measure based classification of medical data for improved classification. Cluster Comput 22 (Suppl 6), 15073–15080 (2019). https://doi.org/10.1007/s10586-018-2498-z
Author Response
The authors answered my comments. It would be great if the authors included some relevant classification reference relating to recent trends (e.g., deep learning)
We agree with your comment. we added the references in Line 498 and 504